# Strong structuring arising from weak cooperative O-H⋯π and C-H⋯O hydrogen bonding in benzene-methanol solution

Camilla Di Mino [1], Andrew G. Seel[2] ✉, Adam J. Clancy [3], Thomas F. Headen [2], Támas Földes[1], Edina Rosta [1], Andrea Sella [3] ✉ & Neal T. Skipper [1] ✉

Weak hydrogen bonds, such as O-H⋯π and C-H⋯O, are thought to direct biochemical assembly, molecular recognition, and chemical selectivity but are seldom observed in solution. We have used neutron diffraction combined with H/D isotopic substitution to obtain a detailed spatial and orientational picture of the structure of benzene-methanol mixtures. Our analysis reveals that methanol fully solvates and surrounds each benzene molecule. The expected O-H⋯π interaction is highly localised and directional, with the methanol hydroxyl bond aligned normal to the aromatic plane and the hydrogen at a distance of 2.30 Å from the ring centroid. Simultaneously, the tendency of methanol to form chain and cyclic motifs in the bulk liquid is manifest in a highly templated solvation structure in the plane of the ring. The methanol molecules surround the benzene so that the O-H bonds are coplanar with the aromatic ring while the oxygens interact with C-H groups through simultaneous bifurcated hydrogen bonds. This demonstrates that weak hydrogen bonding can modulate existing stronger interactions to give rise to highly ordered cooperative structural motifs that persist in the liquid phase.

Hydrogen bonding systems are characterised by short X−H⋯Y inter-actions. Such bonds have traditionally been associated with X = N, O and have a strong electrostatic component arising from the highly dipolar nature of X−H bonds. However, in 1962 Jane Sutor reported the existence of short C-H⋯O contacts in the solid state, a discovery that opened the way for the exploration of a wide variety of weak hydrogen bonding interactions involving less electronegative elements acting as both donors and acceptors[1,2]. Beyond the traditional hydrogen bond-ing interaction, the highly polarised nature of the O-H bond allows the electron rich π system of an aromatic ring to act as a hydrogen bond acceptor. Such O/N/C-H⋯π interactions have been found in the solid state for both small molecule and biomolecule crystals and are now recognised as contributing significantly to molecular recognition and protein folding[3–14].

Spectroscopic methods such as rotational microwave, IR, and NMR are traditionally used to identify such non-conventional interac-tions, including those involving aromatics and the -XH groups of molecules such as water, ammonia, and methanol. NMR in particular has produced advances in identifying weak hydrogen-bonding in crystalline phases via proton-mediated *J* couplings[15,16]. However, detecting such effects experimentally in the liquid phase presents formidable challenges due to rapid molecular reorganisations and competing interactions. In the case of $H_2O$ and $NH_3$ this is further compounded by their extremely low solubility in benzene, making a full characterisation of weak interactions very challenging[13,17–22]. Moreover, while these spectroscopic techniques point towards the existence of such bonding motifs, they do not allow a specific deter-mination of the position of the hydrogens, nor do they reveal details of

[1]Department of Physics and Astronomy, University College London, London WC1E 6BT, UK. [2]ISIS Neutron and Muon Source, Science and Technology Facilities Council, Rutherford Appleton Laboratory, Didcot OX11 0QX, UK. [3]Department of Chemistry, University College London, 20 Gordon Street, London WC1H 0AJ, UK. ✉e-mail: andrew.seel@stfc.ac.uk; a.sella@ucl.ac.uk; n.skipper@ucl.ac.uk

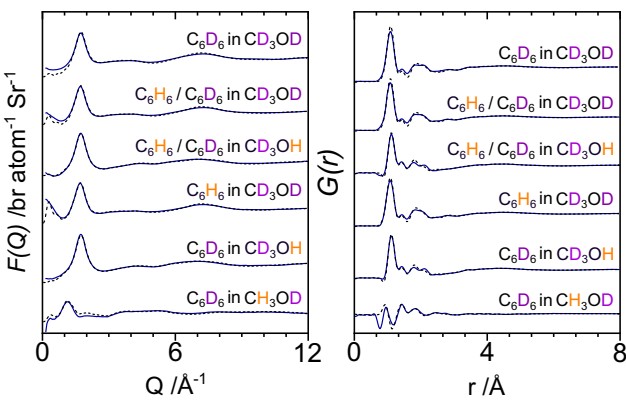

**Fig. 1 | Neutron total structure factors *F(Q)* and total radial distribution functions *G(r)*.** Experimental (solid) and modelled (dashed) neutron diffraction total structure factors (left) and total radial distribution functions (right) for the six isotopically distinct samples. Note how the use of isotope H/D labelling allows us to change the relative contributions from the different atomic sites (see Supplementary Table 1) for full neutron weightings, thereby enabling us to resolve subtle structural effects in the benzene-methanol solution.

the effect of hydrogen bonding on solvation structures in the liquid state. These deficiencies can be addressed by neutron diffraction.

Previous research on bulk liquids via neutron diffraction has included studies of pure aromatic systems, such as benzene, toluene and anisole, which have revealed the presence of π···π, C-H···π, C-H···O weak interactions[23,24]. In the last of these studies, the presence of the electron-donating methoxy group in anisole strongly affects the π cloud while the methyl group of the substituent precludes the possibility of O-H···π bonding.

To understand the underlying structural nature of weak hydrogen bonding interactions in the liquid state, we have therefore explored the archetypal and simplest fully miscible binary system: benzene in methanol. Indeed, ab initio studies on benzene-methanol molecular clusters propose three optimised minimum energy configurations exhibiting: O-H···π, C-H···π and C-H···O weak hydrogen bonding[25]. To answer the key question of which, if any, of these possible configurations are actually present in the liquid state, the current work exploits state-of-the-art neutron diffraction with selective isotopic substitution, supported by Monte Carlo molecular simulations refined against the experimental data. This approach enables us to determine the extent of structural order on a site-by-site basis, including the detailed relative position of hydroxyl groups around benzene. Our results elucidate the precise nature of two dominant non-covalent interactions between the OH group in-plane and out-of-plane from the π system.

## Results and discussion

The experimental neutron diffraction structure factors, *F(Q)*, and EPSR refinements are presented for the six isotopically varied systems alongside their total pair distribution functions, *G(r)*, in Fig. 1. Excellent agreement has been achieved between model and experimental data for each sample dataset; small discrepancies at low-Q are attributed to a residual presence of inelastic and multiple scattering events.

The lack of low-Q scattering signals in the experimental *F(Q)s* of Fig. 1 is, in itself, a clear indication of the absence of benzene-benzene clustering in the 1:19 methanol solution. To investigate this aspect further, Fig. 2 presents the centre-of-mass (CoM)−CoM partial radial distribution functions, $g_{CoM-CoM}(r)$, and coordination numbers, $N_{CoM-CoM}(r)$, for benzene-benzene, methanol-methanol and benzene-methanol interactions (see Supplementary Note 1 for function definitions). Table 1 provides the corresponding peak positions and the average first shell coordination numbers and their respective integration limits. We note first that in our 1:19 mixture, each benzene

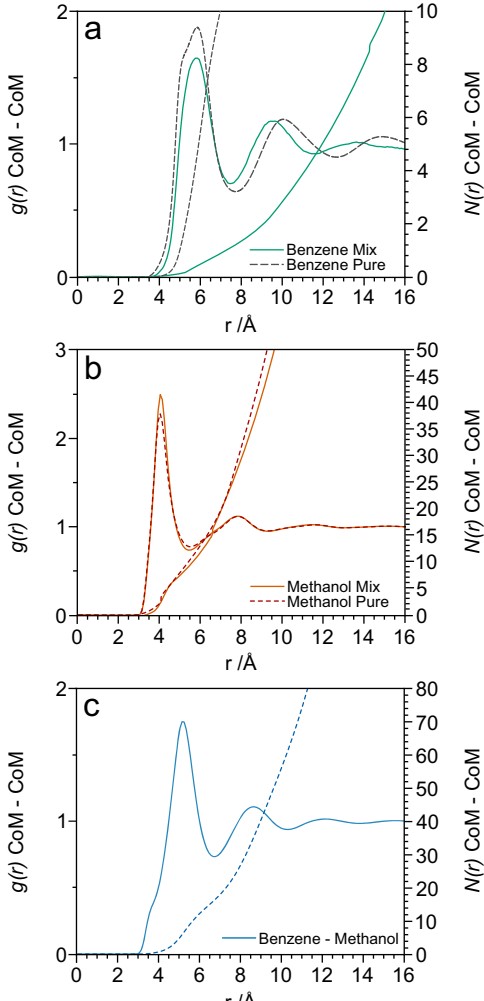

**Fig. 2 | Centre-of-Mass (CoM) intermolecular partial distribution functions for benzene-benzene, methanol-methanol and benzene-methanol.** CoM−CoM partial radial distribution functions $g_{CoM-CoM}(r)$ plotted alongside the coordination numbers $N_{CoM-CoM}(r)$: **a** benzene-benzene interaction in bulk liquid benzene (dashed-line) and in the 1:19 benzene-methanol mixture (solid-line); **b** methanol-methanol interaction in bulk liquid methanol (dashed-line) and in the 1:19 benzene-methanol mixture (solid-line); **c** benzene-methanol interaction in the 1:19 benzene-methanol mixture.

molecule is coordinated to an average of 16.4 methanol molecules (Table 1). In contrast to this, we observe only 1.2 benzene-benzene contacts up to an integration limit of 7.8 Å in the mixture, compared to 12.8 in pure liquid benzene. We conclude that in the 1:19 benzene-methanol mixture, the solvation environment around benzene is dominated by interactions with methanol molecules.

Focusing then on Fig. 2a, it is interesting to note that while the number of benzene-benzene contacts is reduced by more than a factor 10 in the 1:19 mixture compared to the pure liquid, the position of the first peak in $g_{CoM-CoM}(r)$ is almost unaltered (5.88 Å in the bulk and 5.80 Å in the mixture). This indicates that benzene-benzene contacts adopt comparable structural motifs in the two cases (see Supplementary Note 4). The shift of the second benzene-benzene solvation shell to shorter distances (from 10.10 Å in the pure liquid to 9.55 Å in 1:19 methanol solution) indicates that the molecular environment between two second-shell benzenes is constituted by methanol molecules (which are smaller and with higher packing efficiency)[26]. This observation provides further evidence for benzene solubilization. Figure 2b shows that the CoM methanol-methanol structure is robust

**Table 1 | Coordination Numbers $N(r)$ for Benzene-Benzene, Methanol-Methanol and Benzene-Methanol**

|  | 1st peak position /Å Pure / 1:19 Mix | 2nd peak position/Å Pure/1:19 Mix | Integration limit /Å | Coordination number ± 0.1 Pure/mix |
|---|---|---|---|---|
| Benzene–Benzene | 5.88 / 5.80 | 10.10 / 9.55 | 7.8 | 12.8/1.2 |
| Methanol–Methanol | 4.05 / 4.10 | 7.86 / 7.84 | 5.6 | 11.0/10.0 |
| Benzene–Methanol | 5.20 | 8.65 | 6.8 | –/16.4 |

Coordination numbers obtained by integrating the $N_{CoM-CoM}(r)$ up to the first minimum of the CoM—CoM partial distribution functions of Fig. 2. The coordination numbers are reported for pure benzene, pure methanol and the 1:19 benzene-methanol mixture, and indicate the average number of neighbouring molecules within the first solvation shell (see Supplementary Note 1).

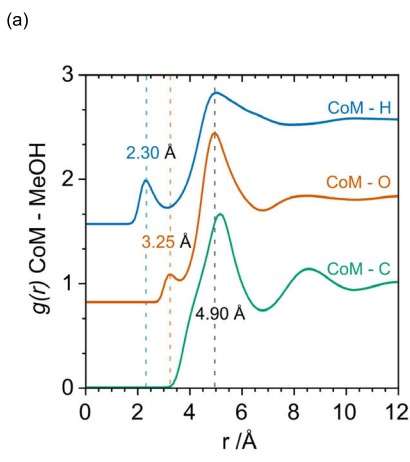
(a)

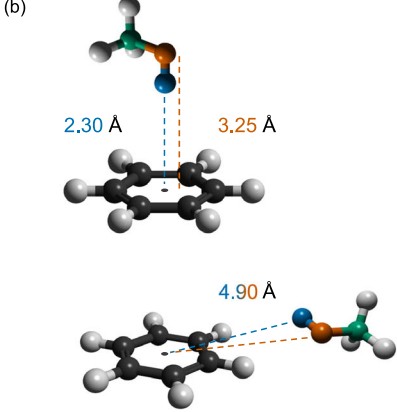
(b)

**Fig. 3 | Intermolecular partial distribution functions between benzene Centre-of-Mass (CoM) and the methanol Hydrogen (H), Oxygen (O) and Carbon (C) sites. a** Benzene CoM−MeOH Partial Distribution Functions: from top to bottom CoM–H(O) blue, CoM−O(H) orange, CoM−C green. **b** A schematic representation of the primary interactions is shown, highlighting the comparable distances between the H/O $g(r)$s peaks and the in-plane position of H/O atoms at further distances.

to the presence of relatively small amounts of benzene, with the first peak at 4.05 Å shifting to 4.10 Å while the coordination number reduces from 11.0 to 10.0 for pure methanol and 1:19 benzene-methanol respectively. The CoM benzene-methanol radial distribution function exhibits clear first and second solvation shells, with peaks at 5.20 and 8.65 Å (Fig. 2c). In addition, there is a shoulder in this function at around 4 Å that points to specific orientation dependent interactions between benzene and methanol. These interactions can be unravelled by interrogating the site-specific correlations that can be extracted by the use of isotopic substitution.

The existence of O-H···π hydrogen bonding is clearly evident from Fig. 3 reporting the benzene centre-of-mass (CoM) - methanol partial distribution functions extracted from the refined structural model for the H, O and C sites of the methanol[27]. The closest approach of the hydrogen to the benzene CoM arises at 2.30 Å as shown in the $g_{CoM-H}(r)$. We conclude that the methanol protons must approach from directly above and below the plane of the benzene ring, as the observed CoM-H distance is too short to allow in-plane approach. The $g_{CoM-O}(r)$ demonstrates that the closest oxygen atom to the benzene ring is found at 3.25 Å. This is 0.95 Å further than the CoM-H distance and therefore precisely within the window of H-O intra-molecular covalent bonding length, implying that the ring CoM···H-O contact is linear[28]. The average coordination number of these hydrogen atoms within a distance of 3.5 Å from the CoM of the benzene is found to be $N_{CoM−H} = 0.59 \pm 0.05$ (see Supplementary Note 1). Since benzene has two π-faces, it can accommodate up to two ring CoM···H-O contacts, and there is therefore the possibility for each molecule to accept 0, 1 or 2 hydrogen bonds with H(-O) donors[3]. To determine the relative likelihoods of these three possible scenarios in solution, we have interrogated the EPSR configurations of our system molecule-by-molecule. This analysis indicates that the probabilities of 0, 1 and 2 hydrogen bonds are 0.51, 0.41 and 0.08 respectively (Fig. 4). The first of these

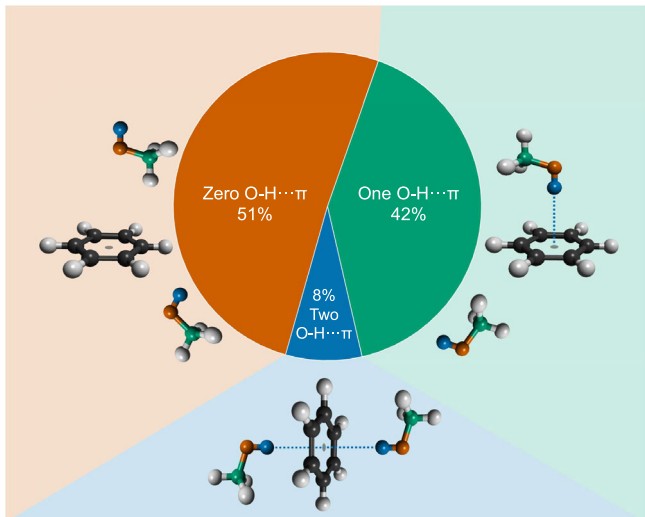

**Fig. 4 | Occurrence of three principal out-of-plane contacts between benzene and methanol.** Probabilities of 0 (Zero, orange), 1 (One, green), and 2 (Two, blue) O-H···π interactions. Benzene can accommodate up to two methanol simultaneously.

values highlights that such CoM···H-O interactions are indeed weak in the liquid state. This then raises the question as to whether the two π-faces of each benzene molecule act independently as hydrogen bond acceptors. To answer this question, we note that the average probability of any π-face accepting an H-bond is given by $P_\pi = N_{CoM−H}/2 = 0.295 \pm 0.025$, and not accepting an H-bond is $\bar{P}_\pi = 1 - P_\pi = 0.705 \pm 0.025$. We can then calculate the probabilities $P_{0,1,2}$ of 0, 1 and 2 H-bonds per benzene molecule, on the assumption

that the two π-faces act independently:

$$P_0 = \bar{P}_\pi \cdot \bar{P}_\pi = 0.497\,,$$
$$P_1 = 2 \cdot \bar{P}_\pi \cdot P_\pi = 0.416\,,$$
$$P_2 = P_\pi \cdot P_\pi = 0.087\,.$$

Comparing with the EPSR molecule-by-molecule observations, we conclude that there is no significant difference between the two methods: this is consistent with the two π-faces acting independently. Our results are therefore consistent with an O-H⋅⋅⋅π interaction of electrostatic nature, where the donor interacts with the acceptor centroid linearly[3].

The second peaks of the hydrogen and oxygen partial distribution functions appear equidistant at 4.9 Å. These distances correspond to both the primary solvation shell in the plane of the benzene ring and the second solvation shells above and below the ring. Importantly the $g_{CoM-C}(r)$ indicates that the methyl groups are found at further distances from the benzene molecule than either hydrogen or oxygen atoms, indicating a strong preference for solvation of the benzene by the hydroxyl groups.

The local structure of methanol around benzene can therefore be partitioned into two distinct environments: in-plane and out-of-plane to the ring. To understand the spatial and orientational arrangement of

methanol in the environments, we can examine the Angular Radial Distribution Functions (ARDFs), the partial distribution functions dependent on the relative orientation of the two molecules, and the Spatial Density Functions (SDFs), the three-dimensional density distribution between two defined species. In particular, in Fig. 5a₁, b₁, c₁ we can determine the relative orientation of the $C_6$ axis of the benzene and the O-H/C-O vectors. Figure 5a₁ shows two sharp peaks at 0° and 180° at ~3 Å distance, confirming that, in absence of steric hindrance from either internal or external components, the O-H approaches the π system directly along the principal axis. The linearity of the interaction is further confirmed in the SDF of Fig. 5a₂.

To characterize the interactions in the plane of the ring, we examine the ARDF for the orientation of the O-H vector of the methanol relative to the molecular plane of benzene. This benzene to methanol O-H ARDF is presented in Fig. 5b₁, and shows a peak centred at 90°, allowing both methanol O and H sites to remain in the benzene plane and at similar distances to the benzene CoM. The refined structure for the solvation of benzene is presented in Fig. 5b₂. The oxygen atom acts here as a bifurcated H-bond acceptor, bonding simultaneously with two hydrogen atoms on the benzene at a distance of 2.9 Å (Supplementary Fig. 3), precluding the presence of a traditional linear C-H⋅⋅⋅O weak hydrogen bond between C-H and O[3]. As a result, the observed configuration of the first benzene in-plane solvation shell is an equatorial 'belt-like' structure around the ring,

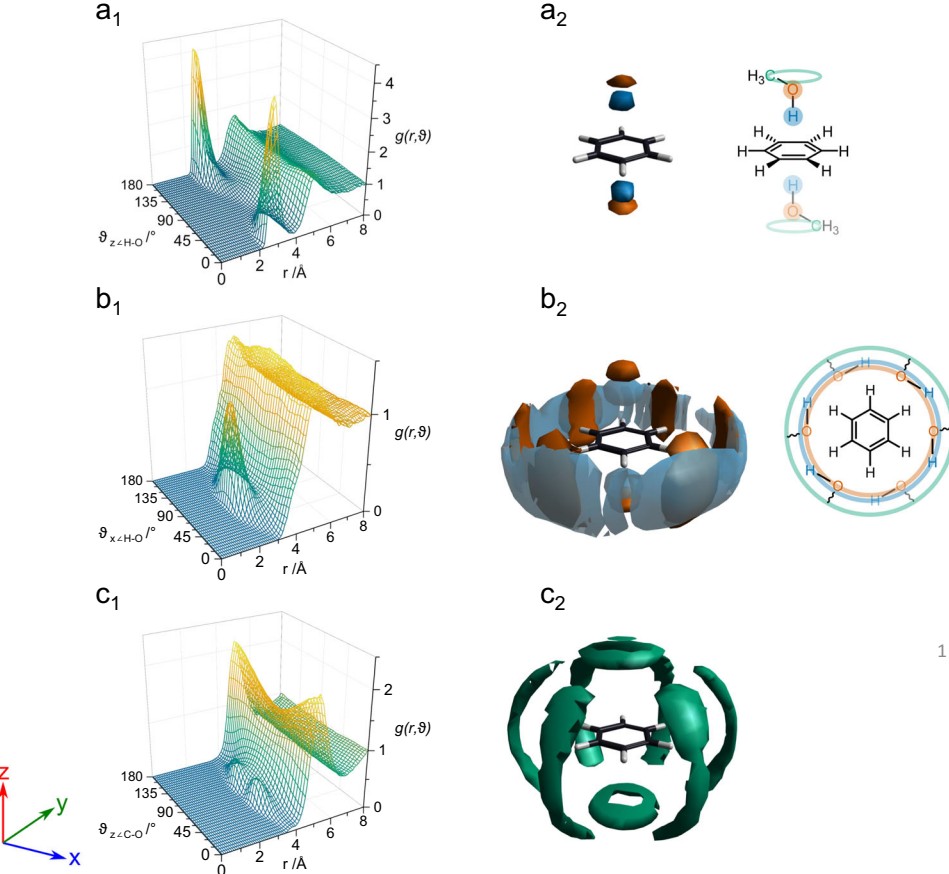

**Fig. 5 | Three-dimensional representation of benzene-methanol contacts: Angular Radial Distribution Functions (ARDFs) and Spatial Density Functions (SDFs) . a₁, b₁, c₁** Angular Radial Distribution Functions (ARDFs) for benzene ring centre -OH methanol, $g(r, \vartheta)$ where r is the distance between the sites and $\vartheta$ is the angle between the normal to the ring plane (z axis) and the O-H (a₁, b₁) and C-O axis (c₁). **a₂, b₂, c₂** Spatial Density Functions (SDFs) showing the most likely positions for molecules in the first coordination shell (see Supplementary Note 1 for function definitions): **a₂** shows the 30% most likely position methanol hydroxyl H (blue) and methanol O (orange) up to 3.1 Å and 3.7 Å, respectively from the benzene CoM; **b₂** shows the 5% most likely position of methanol H and methanol O up to 6.8 Å; from the benzene CoM; **c₂** shows the 10% most likely position of methanol C (green) up to 6.8 Å from the benzene CoM. Note that symmetric features in the SDF reflect the symmetry of benzene and for the latter reason are faded in the schematic of figure. a₂, b₂. Please see Supplementary Fig. 4 for comprehensive axis definitions and Supplementary Note 4 for definitions of the ARDF % likely positions.

consistent with the known tendency of methanol to form long chains both in bulk and in the presence of benzene[29,30]. This belt-like structuring is enhanced by cooperative O-H⋯π and C-H⋯O interactions, where the π-ring acts simultaneously as an H-bond acceptor (O-H⋯π) and H-bond donor (C-H⋯O). This effect has been predicted in ab initio computations and leads to shorter, stronger bonds than for isolated O-H⋯π/C-H⋯O interactions[25]. We note that this stabilising behaviour is not predicted by classical molecular dynamics simulations, nor Monte Carlo simulations in the absence of the refinement from neutron scattering profiles (Supplementary Table 3 and Supplementary Figs. 8 and 9). We have now shown that such interactions play an important role in solution structures[31,32].

Figure 5c$_{1,2}$ shows the most likely position of the methyl groups surrounding the benzene molecule. The ARDFs (Fig. 5c$_1$) show that the relative angle between the C$_6$ axis of the benzene ring and the C-O vector is at 60°–120° which translates to a ring-shaped spatial density above and below the benzene plane in the SDFs of Fig. 5c$_2$. Furthermore, the methyl groups are found around the ring in a complementary position to the O and H sites, in agreement with the belt-like structure, reinforcing the conclusion that solvation of benzene takes place via the hydroxyl groups rather than via methyl-groups. This is clearly an indication of how much more significant dipolar interactions are compared to dispersion in this system.

Comparing our experimentally derived solvation structures with those proposed by earlier ab initio for benzene-methanol clusters, we note that the observed O-H⋯π is directed towards the ring centre, rather than the ring carbons[25]. It is also revealed that, despite not being the minimum energy configuration, the bifurcated coplanar structure is not disfavoured[3,33]. It is worth mentioning that ab initio studies on benzene-methanol clusters include the presence of another favourable configuration, which involves a C-H⋯π weak hydrogen bonding between the methyl hydrogens and the benzene ring[25]. This interaction is absent from the g(r) and ARDFs of Figs. 3 and 5 as it is weaker than the competing O-H⋯π and C-H⋯O interactions present in the current system. In addition, the classic benzene-benzene interactions seen in the bulk liquids, namely for benzene face-to-face stacking at shorter distances and Y-stacking at longer distances, are maintained in the mixture (Supplementary Fig. 5)[24,34].

In summary, neutron diffraction in conjunction with H/D isotopic labelling has been used to reveal the rich hydrogen-bonded structure of benzene-methanol solutions. Our analysis is supported by molecular modelling refined against the experimental data. Our study provides detailed insights into the spatial and orientational configurations that occur in solution. We propose an undiscovered solvation structure for benzene, in which a highly directional O-H⋯π bond sits directly above/below the benzene CoM and aligned normal to the aromatic plane. This motif is strongly localised relative to the ring centre, with well-defined H and O distances of 2.30 Å and 3.25 Å respectively. In addition, we find that methanol forms an equatorial belt around benzene, in which C-H⋯O interactions are bifurcated and the solvent aligns so that O-H bonds are parallel to the aromatic plane. We therefore conclude that, even in the liquid state, weak interactions can give rise to well-defined intermolecular structural motifs.

## Methods

Benzene-methanol solutions have been studied as a function of isotopic composition at a molecular ratio of 1 benzene for every 19 methanol molecules (0.05 mole fraction benzene). At this level of dilution, the solvation of benzene is dominated by interactions with methanol while still providing a measurable contribution to the experimental neutron scattering signal (see Supplementary Fig. 2 and Supplementary Table 1). This allows us to probe subtle interactions, including benzene-methanol O-H⋯π and C-H⋯O hydrogen bonds.

### Experimental

Data have been acquired using the Near and InterMediate Range Order Diffractometer (NIMROD) at the ISIS Neutron and Muon Source (Didcot, UK) across a Q range of 0.05 Å$^{-1}$–50 Å$^{-1}$ [35]. Absolute normalisation, instrument background, multiple and inelastic scattering have been corrected using standard procedures as implemented within the Gudrun package to obtain the total structure factors[36].

All the chemicals, H$_6$ and D$_6$ benzene and D$_1$, D$_3$, D$_4$ methanol, were purchased from Sigma Aldrich with purities ≥99.5%. The anhydrous liquids were used as received to produce the six isotopically distinct 1:19 benzene to methanol mixtures.

**List of samples**
- Fully deuterated benzene C$_6$D$_6$ in CH$_3$OD methanol;
- Fully deuterated benzene C$_6$D$_6$ in CD$_3$OH methanol;
- Fully hydrogenated benzene C$_6$H$_6$ in CD$_3$OD methanol;
- Partially deuterated benzene C$_6$D$_6$/C$_6$H$_6$ in CD$_3$OH methanol;
- Partially deuterated benzene C$_6$D$_6$/C$_6$H$_6$ in CD$_3$OD methanol;
- Fully deuterated benzene C$_6$D$_6$ in CD$_3$OD methanol.

The six samples were inserted into flat-plate null coherent scattering titanium/zirconium cells, with 1 mm sample and wall thicknesses and run for a minimum of 2 h each at 273 K. For data correction and calibration, scattering data were also collected from the empty instrument (with and without the empty sample cell), and an incoherent scattering vanadium standard slab of thickness 3 mm.

### Computational

The extensive use of H/D isotopic substitution has permitted the acquisition of six isotopically distinct data sets. This approach provides strong constraints for structural refinement, with an atomistic model for the liquid system (50 benzene and 950 methanol molecules in a periodically repeated cubic box with sides 41.95 Å) having been refined simultaneously to each experimental dataset using the technique of Empirical Potential Structure Refinement (EPSR) (see Supplementary Note 2: EPSR method)[37]. Specific site-centered Radial Distribution Functions (RDFs), Angular Radial Distribution Functions (ARDFs) and Spatial Density Functions (SDFs) were extracted from our experimentally derived structural models using the dlputils and Aten packages (see Supplementary Note 4 for function definitions)[38,39].

## Data availability

All the processed data generated in this study are provided within the paper, its Supplementary Information file and deposited in the Figshare database[40], available at https://doi.org/10.6084/m9.figshare.23816229.v1. The raw data are available at the ISIS Neutron and Muon Source Data Journal[41,42].

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

## Acknowledgements

The authors acknowledge the UK Science and Technology Facilities Council (STFC) for NIMROD beamtime allocation (https://doi.org/10.5286/ISIS.E.RB1510644), SANDALS beamtime allocation (https://doi.org/10.5286/ISIS.E.RB1900050) and the use of SCARF computational facility for EPSR simulations. Engineering and Physical Sciences Research Council (EPSRC, grant EP/R513143/1) is acknowledged for support of a PhD studentship for CDM. AJC thanks the Ramsay Memorial Fellowship Trust and the Royal Society for funding through the University Research Fellowship scheme (URF\R1\221476, RF\ERE\221017). The authors thank Tristan G. A. Youngs, Christopher A. Howard and Daniele Paoloni for useful discussions.

## Author contributions

C.D.M., N.T.S. and A.G.S. conceived the project. C.D.M., N.T.S. and A.G.S. designed experiments with critical inputs from A.J.C. and T.F.H. N.T.S. and A.G.S. performed experiments. C.D.M., N.T.S. and A.G.S. analysed the results. A.S and T.F.H. validated data interpretation. T.F. and E.R. provided M.D. computational input. C.D.M., A.S. and N.T.S. prepared the paper with input from all authors.

## Competing interests

The authors declare no competing interests.
