## [Peer Review File · Nature Communications]

REVIEWER COMMENTS

Reviewer #1 (Remarks to the Author):

This paper presents a detailed study of the interactions of methanol with benzene at a ratio of 50 benzene to 950 methanol molecules to determine the existence of possible hydrogen-pi bonding interactions in the liquid state. This is an interesting question which is difficult to probe by spectroscopic techniques, and where computational studies have proposed several possible minimum energy configurations for such weak hydrogen bonds. The scattering study has been performed using a standard methodology and provides an impressive number of experimental data sets to constrain the monte carlo modeling used to fit the data. This study provides experimental evidence for two dominant non-covalent interactions between the methanol OH group and the benzene pi system which may help explain reactivity and solubility in this and other systems containing such weak hydrogen bonds. Interestingly standard simulation methods used by the authors on the same system do not pick up the interactions seen in the experimental study, so underestimate important contributions to the liquid structure. The work will therefore be of interest not only to computational chemists but also to those interested in extractions, separations, and reactivity in mixed solvents.

The authors should just clarify a couple of points in their work before publication to assist understanding of their paper.

1. Was there a reason for picking 50 benzene to 950 methanol or is this just a convenient ratio where the benzene molecules should each be fully solvated by methanol? There is a comment in Figure S2 caption which suggests that benzene-benzene interactions maintain their bulk structure in these solutions – that would suggest clustering of the benzenes together, rather than full solubilization by methanol. However figure 5 says that there is only one benzene-benzene interaction in the first solvation shell and thus no evidence of clustering. I'm not sure how these two statements are compatible? Is there evidence of benzene chaining instead?

2. Page 5 line 107-108 the authors state "there is no probabilistic preference for 1 rather than 2 hydrogen bonds" however a few sentences previously they say that "the probabilities of 0, 1 and 2 hydrogen bonds are 0.51, 0.41 and 0.08 respectively". This to me suggests that 1-h bond is more probable than 2, so these two sentences appear to contradict each other. Please can the authors clarify what they mean here?

3. On page 7 line 136 the authors say that "This belt-like effect arises from cooperative O-H...n, C-H...O interactions" when discussing the ring of methanol OH around the perimeter of the benzene ring. It is not clear to me from the text how the O-H...n interactions contribute here. Please could the authors add some further explanation on that point?

Reviewer #2 (Remarks to the Author):

The authors present an interesting study on benzene-methanol correlations in a binary solution using neutron scattering with isotope substitution to locate methanol protons correlated with benzene in a binary solution. The manuscript reports on a very specific structuring of methanol about a benzene (Figure 3). Whereas the neutron data are interesting, they do not appear to support the conclusions, which appear to be largely derived from modeling. Specifically:

There are two OH-methanol shown orthogonal to the benzene ring (Figure 3a2). It does not appear that there are peaks at 6.2 Å (H-H) or 7.4 Å (O-O) in the experimental data to support two correlated methanol to one benzene. The text discusses this point but it is not made clear that the data do not provide the information necessary to distinguish one associated H-O from two. The comment (line 107) that the two faces act independently with no probabilistic preference is questionable based on a wide range of precedent and no justification is made for this assertion. Absent assignable H-H or O-O peaks in the data this conclusion is not justified.

There are no clearly discernible correlation peaks in the experimental data beyond about 3 Å (Figure

1) raising concern about the degree to which the results are dependent on the neutron data. The data do not correspond to the calculated partial RDFs shown in the SI, notably at the longer distances.

The Figure 3 caption states confidence in the model as, for example, "5% most likely position". What exactly does this mean?

The relative concentration of benzene/methanol used in the neutron experiments is not explicitly stated although a 1:19 ratio is referenced in several SI figure captions. Benzene/methanol have a wide miscibility range. Whatever the single concentration used for all the scattering studies, it needs to be provided in the main text along with the rationale for this choice. There is also no discussion about the relevance of the structure determined at this one specific ratio to the wider miscibility region. Since the experiments were obtained at a single ratio there is no evidence for the transferability of the structure seen at this specific ratio to the extended benzene/methanol phase space.

Overall, the broad conclusions presented in this manuscript are not supported by the data presented. The finding of methanol proton centered in the benzene pi-cloud, with a H-O bond perpendicular to the benzene plane appears to be a defensible result that may be appropriately published in a more specialized journal.

Reviewer #3 (Remarks to the Author):

This is a nice study of weak hydrogen bonds using unique capabilities at the ISIS facility, measuring neutron diffraction, comparing H and D substituted molecular systems. This measurement provides detailed insight into spatial and orientational correlations. The results provide an important benchmark to molecular simulations as well as a qualitative understanding of competing molecular interactions, be it O-H ... pi or C-H ... O, dipole interactions, or dispersion interactions. The incremental complexity of the miscible binary system of benzene in methanol is significant and relevant.

Reviewer #4 (Remarks to the Author):

This publication presents an original neutron diffraction study on the benzene-methanol binary mixture, extended by Monte Carlo simulations. The intramolecular orientation of both solvent partners in the liquid is explained presenting their cooperative effect. The paper shows for the first time the high directionality of the O-H...pi bond toward the center of the pi-ring and the highly equatorial structure of the C-H...O bonds relative to benzene.

This study is novel work with a high significance to the field of molecular liquids and binary mixtures. This will give an enhanced insight in weak interactions of the title bonding. The detailed analysis and results are clearly described to guide the reader to these interesting conclusions.

Recommendation: accept with minor changes.

Minor points:

- The authors should give the experimental 1:19 benzene-methanol molar ratio also in the main manuscript.
- The conclusion that 51% probability for zero and 41% for one methanol molecule in the axial position of the pi-face is correlated with the fact that CoM...H-O is a weak interaction (line 105 on p.5), is not clear without defining criteria for a weak bond. This fact could be supported by the information how much methanol molecules surround one benzene molecule on average and in this position.
- From Fig. S5 it is visible that the benzene-benzene distance stays the same for the first solvation shell, but the second solvation shell is contracted upon addition of methanol. This is an important fact, also for the conclusion.
- Can the authors extract the information, how the benzene molecules are located relative to each other (only the distance of ca. 5.5 Angstrom is visible from Fig. S5), the ARDFs could help

interpreting, how methanol is located between two benzene molecules.

- The structures in Fig. 2 are difficult to interpret inside the figure. I suggest to add them aside the graph with additional markings of the distances.

**Reviewer #1 (Remarks to the Author):**

*This paper presents a detailed study of the interactions of methanol with benzene at a ratio*
*of 50 benzene to 950 methanol molecules to determine the existence of possible hydrogen-pi*
*bonding interactions in the liquid state. This is an interesting question which is difficult to*
*probe by spectroscopic techniques, and where computational studies have proposed several*
*possible minimum energy configurations for such weak hydrogen bonds. The scattering*
*study has been performed using a standard methodology and provides an impressive*
*number of experimental data sets to constrain the monte carlo modeling used to fit the*
*data. This study provides experimental evidence for two dominant non-covalent interactions*
*between the methanol OH group and the benzene pi system which may help explain*
*reactivity and solubility in this and other systems containing such weak hydrogen bonds.*
*Interestingly standard simulation methods used by the authors on the same system do not*
*pick up the interactions seen in the experimental study, so underestimate important*
*contributions to the liquid structure. The work will therefore be of interest not only to*
*computational chemists but also to those interested in extractions, separations, and*
*reactivity in mixed solvents.*

**Author reply:**

- • We thank the referee for their evaluation of our work and recommendation.

*The authors should just clarify a couple of points in their work before publication to assist*
*understanding of their paper.*

*Was there a reason for picking 50 benzene to 950 methanol or is this just a convenient ratio*
*where the benzene molecules should each be fully solvated by methanol?*

**Author reply:**

- • As the referee has noted in their summary, the primary aim of this work is to study
weak interactions between methanol and benzene. Hence, we did indeed investigate
a ratio (1:19) where each benzene molecule should be well solvated by methanol. In
addition, at this concentration the neutron scattering contributions from benzene
are measurable and the data therefore allows us to probe, for example, subtle
intermolecular OH $\cdots\pi$ and CH \cdots O hydrogen bonds.

We chose a molecular composition of 50 benzene and 950 methanol (1:19 ratio) for
our EPSR modelling of the data as this system has a cubic box side of 41.95 Å and is
therefore sufficiently large to capture possible solute-solute, solute-solvent and
solvent-solvent intermolecular interactions.

**Suggested changes to the manuscript:**

- • We have added explanatory text on page 4 lines 41 – 45:

“Benzene-methanol solutions have been studied as a function of isotopic
composition at a molecular ratio of 1 benzene for every 19 methanol molecules
(0.05 mole fraction benzene). At this level of dilution, the solvation of benzene is
dominated by interactions with methanol while still providing a measurable
contribution to the experimental neutron scattering signal (see Supporting
Information Table S1). This allows us to probe subtle intermolecular interactions,
including benzene-methanol OH $\cdots\pi$ and CH \cdots O hydrogen bonds.”

- • We have added explanatory text on page S4 lines 87 – 89:
“This model of the system reproduces the experimental composition and density and
is sufficiently large to capture possible solute-solute, solute-solvent and solvent-
solvent intermolecular interactions”.

*There is a comment in Figure S2 caption which suggests that benzene-benzene interactions*
*maintain their bulk structure in these solutions – that would suggest clustering of the*
*benzenes together, rather than full solubilization by methanol. However figure 5 says that*
*there is only one benzene-benzene interaction in the first solvation shell and thus no*
*evidence of clustering. I’m not sure how these two statements are compatible? Is there*
*evidence of benzene chaining instead?*

**Author reply:**

- • We are very grateful to the Reviewer for pointing this out. We agree that our
comment in the caption to Figure S2 (now Figure S3) “Benzene and methanol
maintain their bulk structure in the mixture” is at best confusing, since it might
suggest clustering of benzene in the mixture. This, as stated in reference to Figure S5
(now Figure 2a) and elsewhere, is not the case.

In liquid systems the partial radial distribution functions (PRDFs), such as those
shown in Figures S2 (now S3) and S5 (now 2), are normalized to 1 at large- r . This
definition allows us to compare directly the underlying site-site structure in systems
with different molecular number densities, without recourse to rescaling. However,
to obtain coordination numbers we need to integrate PRDFs, $g_{\alpha\beta}(r)$, according to
Equation S5 (was S8).

In the case of benzene-benzene, the first peaks in $g_{\text{CoM-CoM}}(r)$ are strikingly similar
in the pure liquid and the benzene-methanol mixture. We have augmented this point
to show the benzene-benzene angular radial distribution functions in Figure S5. This
analysis indicates that, where present, the benzene-benzene contacts adopt
comparable structural motifs in the two cases. However, the coordination numbers
differ by more than a factor of ten (12.8 and 1.2 for pure liquid and mixture
respectively). This latter value points to a lack of benzene-benzene clustering in the
mixture. We don’t see evidence for benzene-benzene chaining from the EPSR
configurations, and note that in general for chaining we would expect
$N_{\text{CoM-CoM}}(r) \geq 2$.

**Suggested changes to manuscript:**

- • In the caption for Figure S3 (was Figure S2) we have replaced “Benzene and
methanol maintain their bulk structure in the mixture” with “On mixing, changes to
the underlying site-site functions of benzene and methanol are relatively subtle”.
- • We have merged section S1 “Neutron Diffraction Theory” with S5 “Coordination
Numbers” so that the complementarity of $g_{\alpha\beta}(r)$ and $N_{\alpha\beta}(r)$ are made clear,
including the role of number density ρ_{β} in the latter. In addition, in this section on
page S2 lines 33-36 we have added:
“relative probability”

“In liquid systems, where there is no long-range order, the $g_{\alpha\beta}(r)$ in equation S3
therefore have an asymptote of 1 at large- r and give important site-specific
structural information of the system.”

• We have merged Figures S5, S6 and S7 of the SI to give new Figure 2 of the main
text. Figure 2a, b and c show the CoM – CoM partial radial distribution functions
$g_{\text{CoM-CoM}}(r)$ plotted alongside the coordination numbers $N_{\text{CoM-CoM}}(r)$ for
benzene-benzene, methanol-methanol, benzene-methanol in the bulk and in the
1:19 mixture. The benzene-benzene $N_{\text{CoM-CoM}}(r)$ has been added to Figure 2a to
highlight the point made by the referee.

• We have added Table 1 to the main text containing the $g_{\text{CoM-CoM}}(r)$ first peak
positions, coordination numbers and integration limits corresponding to Figure 2.

• We have added substantial explanatory text regarding benzene-benzene interactions
and comparison between pure liquids and the mixture on page 5 lines 81-109:

“The lack of low- Q scattering signals in the experimental $F(Q)$ s of Figure 1 is, in itself,
a clear indication of the absence of benzene-benzene clustering in the 1:19 methanol
solution. To investigate this aspect further, Figure 2 presents the centre-of-mass
(CoM) – CoM partial radial distribution functions, $g_{\text{CoM-CoM}}(r)$, and coordination
numbers, $N_{\text{CoM-CoM}}(r)$, for benzene-benzene, methanol-methanol and benzene-
methanol interactions (see Supporting Information Section S1 for function
definitions). Table 1 provides the corresponding peak positions and the average first
shell coordination numbers and their respective integration limits. We note first that
in our 1:19 mixture, each benzene molecule is coordinated to an average of 16.4
methanol molecules (Table 1). In contrast to this, we observe only 1.2 benzene-
benzene contacts up to an integration limit of 7.8 Å in the mixture, compared to 12.8
in pure liquid benzene. We conclude that in the 1:19 benzene-methanol mixture, the
solvation environment around benzene is dominated by interactions with methanol
molecules.

Focusing then on Figure 2a, it is interesting to note that while the number of
benzene-benzene contacts is reduced by more than a factor 10 in the 1:19 mixture
compared to the pure liquid, the position of the first peak in $g_{\text{CoM-CoM}}(r)$ is almost
unaltered (5.88 Å in the bulk and 5.80 Å in the mixture). This indicates that benzene-
benzene contacts adopt comparable structural motifs in the two cases (please see
also Supporting Information section S4). The shift of the second benzene-benzene
solvation shell to shorter distances (from 10.10 Å in the pure liquid to 9.55 Å in 1:19
methanol solution) indicates that the molecular environment between two second-
shell benzenes is constituted by methanol molecules (which are smaller and allow
higher packing efficiency)³¹. This observation provides further evidence for benzene
solubilization. Figure 2b shows that the CoM methanol-methanol structure is robust
to the presence of relatively small amounts of benzene, with the first peak at 4.05 Å
shifting to 4.10 Å while the coordination number reduces from 11.0 to 10.0 for pure
methanol and 1:19 benzene-methanol respectively. The CoM benzene-methanol
radial distribution function exhibits clear first and second solvation shells, with peaks
at 5.20 and 8.65 Å (Figure 2c). In addition, there is a shoulder in this function at
around 4 Å that points to specific orientation dependent interactions between

benzene and methanol. These interactions can be unraveled by interrogating the
site-specific correlations that can be extracted by the use of isotopic substitution.”

• We have added Figure S5 which shows the Angular Radial Distribution Functions
(ARDFs) for benzene-benzene neighbours in the bulk liquid and mixture. This figure
quantifies the structural similarities and differences in the two cases.

• We have added lines page S10 152 – 160:

“Figure S5 presents the ARDFs of the relative orientation of the C6 axis of two
distinct benzene molecules in bulk benzene and in the 1:19 benzene-methanol
mixture. The two ARDFs are similar as they present a first weak peak at 0 ° and 180°,
and, at further distances, they show a sharp well-defined peak at 90° degrees
indicating parallel-displaced and Y displacement respectively. The intensities of the
peaks indicate that the benzene-benzene interaction is much weaker in the
methanol mixture than in the pure liquid, and this observation is consistent with the
information extracted from the partial distribution functions of Figure 2 and the
respective benzene-benzene coordination number of 1.2 (Table 1).”

2. Page 5 line 107-108 the authors state “there is no probabilistic preference for 1 rather
than 2 hydrogen bonds” however a few sentences previously they say that “the probabilities
of 0, 1 and 2 hydrogen bonds are 0.51, 0.41 and 0.08 respectively”. This to me suggests that
1-h bond is more probable than 2, so these two sentences appear to contradict each other.
Please can the authors clarify what they mean here?

**Author reply:**

• We apologize for the unclear/contradictory phrasing here.

As the Reviewer notes, molecule-by-molecule analysis of the EPSR configurations of
our system yields probabilities of 0.51, 0.41 and 0.08 for 0, 1 and 2 benzene-
methanol OH··· π hydrogen bonds respectively. So, 1 H-bond is indeed more probable
than 2, as we would expect for these weak interactions.

However, we then wanted to answer the question as to whether these observed
probabilities are consistent with the two π -faces of each benzene acting
independently as H-bond acceptors. Here we need to provide details and
explanations of the calculations.

On page 7 line 120 we note that the relevant methanol-H to benzene-CoM
coordination number is $N_{\text{CoM-H}} = 0.59 \pm 0.05$. The overall average probability of
any π -face accepting an H-bond is therefore $P_{\pi} = N_{\text{CoM-H}}/2 = 0.295 \pm 0.025$, and
not accepting $\bar{P}_{\pi} = 1 - P_{\pi} = 0.705 \pm 0.025$. We can then calculate the probabilities
$P_{0,1,2}$ of 0, 1 and 2 H-bonds per benzene molecule, on the assumption that the two
π -faces act independently:

$$P_0 = \bar{P}_{\pi} \cdot \bar{P}_{\pi} = 0.497,$$

$$P_1 = 2 \cdot \bar{P}_{\pi} \cdot P_{\pi} = 0.416,$$

$$P_2 = P_{\pi} \cdot P_{\pi} = 0.087.$$

Comparing with the EPSR observations, we conclude that there is no significant
difference between the two methods: this is consistent with the two π -faces acting
independently.

**Suggested changes to manuscript:**

• We have added Figure 4 in the main text showing the probabilities of 0, 1, and 2 O-
H $\cdots\pi$ interactions in benzene/methanol solution, along with illustrative molecular
snapshots.

• We have added page 7 line 130-139:

“To answer this question, we note that the average probability of any π -face
accepting an H-bond is given by $P_\pi = N_{\text{CoM-H}}/2 = 0.295 \pm 0.035$, and not
accepting an H-bond is $\bar{P}_\pi = 1 - P_\pi = 0.705 \pm 0.035$. We can then calculate the
probabilities $P_{0,1,2}$ of 0, 1 and 2 H-bonds per benzene molecule, on the assumption
that the two π -faces act independently:

$$P_0 = \bar{P}_\pi \cdot \bar{P}_\pi = 0.497,$$

$$P_1 = 2 \cdot \bar{P}_\pi \cdot P_\pi = 0.416,$$

$$P_2 = P_\pi \cdot P_\pi = 0.087.$$

Comparing with the EPSR molecule-by-molecule observations, we conclude that
there is no significant difference between the two methods: this is consistent with
the two π -faces acting independently”.

• “We have deleted page 7 line 120: “(there is no probabilistic preference for 1 rather
than 2 hydrogen bonds)”

• We have added page 7 line 125: “(Figure 4)”

3. On page 7 line 136 the authors say that “This belt-like effect arises from cooperative O-
H $\cdots\pi$, C-H \cdots O interactions” when discussing the ring of methanol OH around the perimeter of
the benzene ring. It is not clear to me from the text how the O-H $\cdots\pi$ interactions contribute
here. Please could the authors add some further explanation on that point?

**Author reply:**

• We have added further explanation and amendments on this point, which we hope
will clarify. The effect of cooperation can be seen from the shortened nature of both
O-H $\cdots\pi$ and C-H \cdots O interactions versus classical simulations, including our MC
simulations undertaken with and without the neutron scattering-derived empirical
refinement (Table S3). This shows that interactions where the π -ring acts
simultaneously as an H-bond acceptor (O-H $\cdots\pi$) and H-bond donor (C-H \cdots O) reinforce
each other. We have rephrased this section in the main text to highlight the mutual
nature of the cooperation, including highlighting previous computational literature
which explores the effect in greater detail.

**Suggested changes to manuscript:**

• We have added/amended page 10 lines 168 - 171 in the main text

“This belt-like structuring is enhanced by cooperative O-H \cdots π and C-H \cdots O
interactions, where the π -ring acts simultaneously as an H-bond acceptor (O-H \cdots π)
and H-bond donor (C-H \cdots O). This effect has been predicted in *ab initio* computations
and leads to shorter, stronger bonds than for isolated O-H \cdots π /C-H \cdots O interactions.²⁵”

- We have added a reference to Table S3 on page 10 line 174.
 - We have removed from line 171 “first identified in crystalline phases”

**Reviewer #2 (Remarks to the Author):**

*The authors present an interesting study on benzene-methanol correlations in a binary*
*solution using neutron scattering with isotope substitution to locate methanol protons*
*correlated with benzene in a binary solution. The manuscript reports on a very specific*
*structuring of methanol about a benzene (Figure 3). Whereas the neutron data are*
*interesting, they do not appear to support the conclusions, which appear to be largely*
*derived from modeling.*

**Author reply:**

• We will respond in detail to the Reviewers specific points shortly, but in general we
would like to note that our EPSR refinement is highly constrained by the six
isotopically distinct data sets, each with very different neutron scattering weightings
(see table S1). Further, we directly demonstrate the impact and influence of the
experimental data on our conclusions by modelling with and without neutron
scattering constraint.

*Specifically:*

*There are two OH-methanol shown orthogonal to the benzene ring (Figure 3a2). It does not*
*appear that there are peaks at 6.2 Å (H-H) or 7.4 Å (O-O) in the experimental data to support*
*two correlated methanol to one benzene. The text discusses this point but it is not made*
*clear that the data do not provide the information necessary to distinguish one associated H-*
*O from two. The comment (line 107) that the two faces act independently with no*
*probabilistic preference is questionable based on a wide range of precedent and no*
*justification is made for this assertion. Absent assignable H-H or O-O peaks in the data this*
*conclusion is not justified.*

**Author reply:**

• We are very grateful to the Reviewer for raising this point: we agree that Figure 3a2
will cause confusion, as it shows two OH-methanol interactions when this motif is in
fact rather rare (occurring for only ~8% of benzene molecules). We now show only
one methanol in Figure 3 (was 2), provide a new Figure 4 with probabilities and
example configurations, and have ghosted the lower molecule in Figure 5a2 (was
Figure 3a2). Our original intention with Figure 3a2 was to indicate that the two π -
faces are, a priori, equivalent.

• With regards to the 2-methanol motif now shown in Figure 4 (Bi, 2), we note that
this specific correlation occurs for only 8% of benzene molecules. As such, we would
not expect the through-ring H-H and O-O features to be significant, for example in
the partial radial distribution functions shown in Figure S3. Based on Figure 3 (as 2),
we note also that the relevant H-H and O-O distances would be 4.6 and 6.5 Å (rather
than 6.2 Å and 7.4 Å).

• Concerning our comment on line 107 about independent faces, we agree here that
our text was insufficient and unclear. We now explain that molecule-by-molecule
interrogation of our EPSR configurations leads to probabilities of 51%, 41% and 8%
for 0, 1, and 2 O-H $\cdots\pi$ interactions in benzene/methanol solution (Figure 4). We then
compare these values with those obtained from the average coordination number
$N_{\text{CoM-H}} = 0.59 \pm 0.05$ (obtained by integrating $g_{\text{CoM-H}}$ shown in Figure 3), if we

assume that the two faces are independent. The overall average probability of any π -
face accepting an H-bond is therefore $P_\pi = N_{\text{CoM-H}}/2 = 0.295 \pm 0.025$, and not
accepting $\bar{P}_\pi = 1 - P_\pi = 0.705 \pm 0.025$. We can then calculate the probabilities
$P_{0,1,2}$ of 0, 1 and 2 H-bonds per benzene molecule, on the assumption that the two
π -faces act independently:

$$260 \quad P_0 = \bar{P}_\pi \cdot \bar{P}_\pi = 0.497,$$

$$261 \quad P_1 = 2 \cdot \bar{P}_\pi \cdot P_\pi = 0.416,$$

$$262 \quad P_2 = P_\pi \cdot P_\pi = 0.087.$$

Comparing with the EPSR observations, we conclude that there is no significant
difference between the two methods: this is consistent with the two π -faces acting
independently.

**Suggested changes to manuscript:**

• We have added Figure 4 in the main text showing the probabilities of 0, 1, and 2 O-
H $\cdots\pi$ interactions in benzene/methanol solution and representative structures.

• We have added page 8 line 130-139:

“To answer this question, we note that the average probability of any π -face
accepting an H-bond is given by $P_\pi = N_{\text{CoM-H}}/2 = 0.295 \pm 0.025$, and not
accepting an H-bond is $\bar{P}_\pi = 1 - P_\pi = 0.705 \pm 0.025$. We can then calculate the
probabilities $P_{0,1,2}$ of 0, 1 and 2 H-bonds per benzene molecule, on the assumption
that the two π -faces act independently:

$$275 \quad P_0 = \bar{P}_\pi \cdot \bar{P}_\pi = 0.497,$$

$$276 \quad P_1 = 2 \cdot \bar{P}_\pi \cdot P_\pi = 0.416,$$

$$277 \quad P_2 = P_\pi \cdot P_\pi = 0.087.$$

Comparing with the EPSR molecule-by-molecule observations, we conclude that
there is no significant difference between the two methods: this is consistent with
the two π -faces acting independently”.

• We have deleted page 8 line 139: “(there is no probabilistic preference for 1 rather
than 2 hydrogen bonds)”

• We have added page 7 line 125: “(Figure 4)”

• We have modified Figure 5a₂ (was 3a₂). One of the two OH to π sketches is now
faded.

• Figure 5 (was 3) caption, line 264: “Note that symmetric features in the SDFs reflect
the symmetry of benzene”.

*There are no clearly discernible correlation peaks in the experimental data beyond about 3 Å*
*(Figure 1) raising concern about the degree to which the results are dependent on the*
*neutron data. The data do not correspond to the calculated partial RDFs shown in the SI,*
*notably at the longer distances.*

**Author reply:**

• We thank the Reviewer for this observation. We agree that Figure 1 is not clear
regarding longer-range intermolecular features: this is due to the presence of very

sharp intramolecular peaks at shorter distances. We have addressed this issue in two
ways.

First, in Figure S2 we now show the total RDFs and EPSR refinement over the range 2
298 – 10 Å. On this scale there are indeed clear features beyond 3 Å, which are very well
fitted by our EPSR refinement.

Second, we have moved the CoM-CoM partial correlation functions to Figure 2.
These show clear intermolecular correlations at larger distances.

**Suggested changes to manuscript:**

- • Figure S2 is included to show the total radial distribution functions and EPSR
refinements over the range 2 – 10 Å, set on an appropriate scale to view longer-
range correlations.
- • We have moved and augmented Figure S5 to Figure 2, where we compare CoM-CoM
partial RDFs, including longer-range intermolecular interactions.

*The Figure 3 caption states confidence in the model as, for example, “5% most likely*
*position”. What exactly does this mean?*

**Author reply:**

- • We apologies for the lack of clarity on this point. We have modified the text and
added to the Supporting Information to explain the % most likely positions. In the
Supporting information we have also provided examples of additional SDFs as a
function of % to represent the local coordination spatial distribution more fully.

**Suggested changes to manuscript:**

- • Caption for Figure 5 (was 3) we have added:
“Please see Supporting Information Figure S4 for comprehensive axes definitions and
section S4 for definitions of the ARDF % likely positions”.
- • We have added line 163 – 166 to Supporting Information:
“The 3D grid associated with the SDFs is plotted by filling in the voxels from most
probable downwards and this process is stopped when cumulative probability
reaches a set percentage for the volume over the SDFs were calculated. In the case
of the SDFs of Figure 5 a2, b2 and c2 main text the set percentage is 30%, 5% and
10% respectively”
- • We have added Figure S6 showing the H and O spatial densities up to 3 Å at different
percentages: 10, 30, 50 and 70%.
- • We have added Figure S7 showing the H and O spatial densities up to 6.8 Å at
different percentages: 5, 10, 30%.

*The relative concentration of benzene/methanol used in the neutron experiments is not*
*explicitly stated although a 1:19 ratio is referenced in several SI figure captions.*
*Benzene/methanol have a wide miscibility range. Whatever the single concentration used for*
*all the scattering studies, it needs to be provided in the main text along with the rationale for*
*this choice. There is also no discussion about the relevance of the structure determined at*
*this one specific ratio to the wider miscibility region. Since the experiments were obtained at*

*a single ratio there is no evidence for the transferability of the structure seen at this specific*
*ratio to the extended benzene/methanol phase space.*

**Author reply:**

• We thank the Reviewer for their comment regarding the concentration and we
apologise for this omission. To rectify this issue, we have moved the Experimental
Methods section from the Supporting Information to the main article as section 2,
and have added text to explain the rationale for the choice of 1:19 (0.05 mole
fraction benzene).

The primary aim of this work is to study weak interactions between methanol and
benzene. Hence, we investigated a concentration where each benzene molecule
should be well solvated by methanol. In addition, at the chosen concentration the
neutron scattering contributions centred on benzene (Table S1) are measurable. The
data therefore allows us to probe, for example, subtle intermolecular OH \cdots π and
CH \cdots O hydrogen bonds. At this concentration, the data also give information about
benzene-benzene (and of course methanol-methanol) interactions, though this is a
secondary aim.

• With regards to transferability to the wider miscibility region, we have now noted
that each benzene molecule is coordinated by 16.4 methanol (and only 1.2 benzene).
The solvation environment around benzene is therefore dominated by interactions
with methanol molecules, are for this reason we are confident that our conclusions
are relevant to the more dilute (low benzene concentration) regime in general.

We have also compared benzene-benzene (and methanol-methanol) partial RDFs in
the bulk liquids and 1:19 mixture and note that these functions are strikingly similar.
Moreover, we have calculated the Angular Radial Distribution Functions (ARDFs)
showing the relative orientation of benzene-benzene in the first coordination shell.
We note that in 1:19 solution these interactions are dominated by the same motifs
as in the bulk liquids (parallel displaced and perpendicular “Y”). Hence, we think
there is also some transferability to the more concentrated regime.

**Suggested changes to manuscript:**

• We have moved section S1 “Experimental Method” to form section 2 of the main
text.

• We have added explanatory text regarding the rationale on page 4 lines 41 – 45:

“Benzene-methanol solutions have been studied as a function of isotopic
composition at a molecular ratio of 1 benzene for every 19 methanol molecules
(0.05 mole fraction benzene). At this level of dilution, the solvation of benzene is
dominated by interactions with methanol while still providing a measurable
contribution to the experimental neutron scattering signal (see Supporting
Information Table S1). This allows us to probe subtle intermolecular interactions,
including benzene-methanol OH \cdots π and CH \cdots O hydrogen bonds.”

• We have merged Figures S5, S6 and S7 of the SI to give new Figure 2 of the main
text. Figure 2a, b and c show the CoM – CoM partial radial distribution functions
$g_{\text{CoM-CoM}}(r)$ plotted alongside the coordination numbers $N_{\text{CoM-CoM}}(r)$ for

benzene-benzene, methanol-methanol, benzene-methanol in the bulk and in the
1:19 mixture.

• We have added Table 1 to the main text containing the $g_{\text{CoM-CoM}}(r)$ first peak
positions, coordination numbers and integration limits corresponding to Figure 2.
This shows that the benzene-methanol coordination number is 16.4.

• We have added text that provides justification for the composition on page 6 lines
81-92:

“The lack of low- Q scattering signals in the experimental $F(Q)$ s of Figure 1 is, in itself,
a clear indication of the absence of benzene-benzene clustering in the 1:19 methanol
solution. To investigate this aspect further, Figure 2 presents the centre-of-mass
(CoM) – CoM partial radial distribution functions, $g_{\text{CoM-CoM}}(r)$, and coordination
numbers, $N_{\text{CoM-CoM}}(r)$, for benzene-benzene, methanol-methanol and benzene-
methanol interactions (see Supporting Information Section S1 for function
definitions). Table 1 provides the corresponding peak positions and the average first
shell coordination numbers and their respective integration limits. We note first that
in our 1:19 mixture, each benzene molecule is coordinated to an average of 16.4
methanol molecules (Table 1). In contrast to this, we observe only 1.2 benzene-
benzene contacts up to an integration limit of 7.8 Å in the mixture, compared to 12.8
in pure liquid benzene. We conclude that in the 1:19 benzene-methanol mixture, the
solvation environment around benzene is dominated by interactions with methanol
molecules.

*Overall, the broad conclusions presented in this manuscript are not supported by the data*
*presented. The finding of methanol proton centered in the benzene pi-cloud, with a H-O*
*bond perpendicular to the benzene plane appears to be a defensible result that may be*
*appropriately published in a more specialized journal.*

**Author reply:**

• The broad conclusions of our manuscript are that neutron diffraction in conjunction
with H/D isotopic labelling can be used to reveal the rich hydrogen-bonded structure
of benzene-methanol solutions. Specifically, a highly directional O-H \cdots π bond sits
directly above/below the benzene CoM and aligned normal to the aromatic plane. In
addition, we find that methanol forms an equatorial belt around benzene, in which
C-H \cdots O interactions are bifurcated and the solvent aligns so that O-H bonds are
parallel to the aromatic plane. We appreciate very much the comments of the
reviewer, and we hope that in addressing their comments and modifying the
manuscript accordingly, they can be persuaded that the conclusions are supported
by the data and that the paper is suitable for publication in Nature Communications.

**Reviewer #3 (Remarks to the Author):**

*This is a nice study of weak hydrogen bonds using unique capabilities at the ISIS facility,*
*measuring neutron diffraction, comparing H and D substituted molecular systems. This*
*measurement provides detailed insight into spatial and orientational correlations. The*
*results provide an important benchmark to molecular simulations as well as a qualitative*
*understanding of competing molecular interactions, be it O-H ... pi or C-H ... O, dipole*
*interactions, or dispersion interactions. The incremental complexity of the miscible binary*
*system of benzene in methanol is significant and relevant.*

**Author reply:**

- • We thank the referee for their evaluation of our manuscript and recommendation.

**Reviewer #4 (Remarks to the Author):**

*This publication presents an original neutron diffraction study on the benzene-methanol*
*binary mixture, extended by Monte Carlo simulations. The intramolecular orientation of both*
*solvent partners in the liquid is explained presenting their cooperative effect. The paper*
*shows for the first time the high directionality of the O-H... π bond toward the center of the*
*π -ring and the highly equatorial structure of the C-H...O bonds relative to benzene.*

*This study is novel work with a high significance to the field of molecular liquids and binary*
*mixtures. This will give a enhanced insight in weak interactions of the title bonding. The*
*detailed analysis and results are clearly described to guide the reader to this interesting*
*conclusions.*

*Recommendation: accept with minor changes.*

**Author reply:**

 - We thank the referee for their evaluation of our manuscript and recommendation.

*Minor points:*

*The authors should give the experimental 1:19 benzene-methanol molar ratio also in the*
*main manuscript.*

**Author reply:**

 - We agree with the referee that the experimental molecular ratio should be in the

445 main text and we apologise for the omission.

**Suggested changes to manuscript:**

 - We have moved section S1 “Experimental Method” to form section 2 of the main

text.

 - We have added explanatory text on page 4 lines 41 – 45:

“Benzene-methanol solutions have been studied as a function of isotopic
composition at a molecular ratio of 1 benzene for every 19 methanol molecules (0.05
mole fraction benzene). At this level of dilution, the solvation of benzene is
dominated by interactions with methanol while still providing a measurable
contribution to the experimental neutron scattering signal (see Supporting
Information Table S1). This allows us to probe subtle intermolecular interactions,
including benzene-methanol OH... π and CH...O hydrogen bonds.”

*The conclusion that 51% probability for zero and 41% for one methanol molecule in the axial*
*position of the π -face is correlated with the fact that CoM...H-O is a weak interaction (line*
*105 on p.5), is not clear without defining criteria for a weak bond. This fact could be*
*supported by the information how much methanol molecules surround one benzene*
*molecule on average and in this position.*

**Author reply:**

 - We agree that there is a need to support the weak nature of the interaction, and

464 that the coordination numbers are suitable in this context (particularly that for each

465 π -face the CoM-H coordination number is only 0.295.

**Suggested changes to the manuscript:**

- • On page 6 line 88, the average coordination number of methanol around benzene is
given as 16.4, and from page 7 line 120 the specific OH... π coordination number is
$N_{\text{CoM-H}} = 0.59$ (and for each π -face $P_{\pi} = N_{\text{CoM-H}}/2 = 0.295$).
- • We have merged Figures S5, S6 and S7 of the SI to give new Figure 2 of the main
text. Figure 2a, b and c show the CoM – CoM partial radial distribution functions
$g_{\text{CoM-CoM}}(r)$ plotted alongside the coordination numbers $N_{\text{CoM-CoM}}(r)$ for
benzene-benzene, methanol-methanol, benzene-methanol in the bulk and in the
1:19 mixture.
- • We have added Table 1 to the main text containing the $g_{\text{CoM-CoM}}(r)$ first peak
positions, coordination numbers and integration limits corresponding to Figure 2.

*From Fig. S5 it is visible that the benzene-benzene distance stays the same for the first*
*solvation shell, but the second solvation shell is contracted upon addition of methanol. This is*
*an important fact, also for the conclusion.*

**Author reply:**

- • We thank the Reviewer for this observation, and we agree that the second solvation
shell of benzene-benzene $g_{\text{CoM-CoM}}(r)$ contracts because the space between two
benzene molecules is occupied by methanol molecules, which are smaller and with
higher packing efficiency (H-bonds instead of π - π interactions) than benzene. This
observation further confirms the individualization of benzene molecules in our
system and has been incorporated into the manuscript.

**Suggested changes to manuscript:**

- • We have added page 6 lines 97-100:
“The shift of the second benzene-benzene solvation shell to shorter distances (from
10.10 Å in the pure liquid to 9.55 Å in 1:19 methanol solution) indicates that the
molecular environment between two second-shell benzenes is constituted by
methanol molecules (which are smaller and with higher packing efficiency).³¹”

*Can the authors extract the information, how the benzene molecules are located relative to*
*each other (only the distance of ca. 5.5 Angstrom is visible from Fig. S5), the ARDFs could*
*help interpreting, how methanol is located between two benzene molecules.*

**Author reply:**

- • Yes - we have calculated the benzene-benzene ARDFs for both the bulk liquid (using
the data set of reference 31) and our 1:19 solution. These are shown in the new
Figure S5. These indicate that the two ARDFs are very similar, as they present a first
weak peak at 0° and 180° due to parallel-displaced motifs, and, at further distances,
they show a sharp well-defined peak at 90° indicating the perpendicular “Y” motif.

**Suggested changes to manuscript:**

- • We have added to section S5 Figure S5 showing the Angular Radial Distribution
Functions ARDFs for pure benzene-benzene and benzene-benzene in the 1:19
mixture with methanol.
- • We have added lines page S10 152 – 160:
“Figure S5 presents the ARDFs of the relative orientation of the C6 axis of two
distinct benzene molecules in bulk benzene and in the 1:19 benzene-methanol
mixture. The two ARDFs are similar as they present a first weak peak at 0 ° and 180°,
and, at further distances, they show a sharp well-defined peak at 90° degrees
indicating parallel-displaced and Y displacement respectively. The intensities of the
peaks indicate that the benzene-benzene interaction is much weaker in the
methanol mixture than in the pure liquid, and this observation is consistent with the
information extracted from the partial distribution functions of Figure 2 and the
respective benzene-benzene coordination number of 1.2 (Table 1).”

*The structures in Fig. 2 are difficult to interpret inside the figure. I suggest to add them aside*
*the graph with additional markings of the distances.*

**Author reply:**

- • We apologise for the lack of clarity.

**Suggested changes to manuscript:**

- • We have redrawn Figure 2 (Figure 3 of the current manuscript).

REVIEWERS' COMMENTS

Reviewer #1 (Remarks to the Author):

the authors appear to have addressed all my concerns and comments so the paper should now be able to be published.

Reviewer #4 (Remarks to the Author):

The authors have answered all comments satisfactorily.
With the detailed answers, the manuscript has improved and can be published.